# The Duck *RXRA* Gene Promotes Adipogenesis and Correlates with Feed Efficiency

**DOI:** 10.3390/ani13040680

**Published:** 2023-02-15

**Authors:** Ziyi Pan, Xuewen Li, Dongsheng Wu, Xingyong Chen, Cheng Zhang, Sihua Jin, Zhaoyu Geng

**Affiliations:** College of Animal Science and Technology, Anhui Agricultural University, Hefei 230036, China

**Keywords:** ducks, *RXRA*, adipogenesis, SNP, feed efficiency

## Abstract

**Simple Summary:**

Avoiding feed energy waste and breeding efficient growth ducks have become the main goals of duck breeders due to the increase in feed costs. The feed efficiency of ducks is closely related to adipogenesis. The purpose of this study was to explore the effect of the *RXRA* gene on duck preadipocytes and provide molecular markers for breeding ducks for feed efficiency. The results of this study showed that *RXRA* promoted fat accumulation in duck preadipocytes, and the mutation site of *RXRA* was significantly related to duck feed efficiency.

**Abstract:**

Background: The accumulation of fat in ducks is the main cause of low feed efficiency and metabolic diseases in ducks. Retinoic acid X receptor alpha (*RXRA*) is a member of the nuclear receptor superfamily involved in lipid, glucose, energy, and hormone metabolism. The effect of the *RXRA* gene on lipid metabolism in duck preadipocytes (DPACs) and the relationship between SNPs and the feed efficiency traits of ducks are unclear. Methods: qRT-PCR and Western blotting analyses were used to detect changes in mRNA and protein in cells. Intracellular triglycerides (TGs) were detected using an ELISA kit. A general linear model analysis was used to determine the association between *RXRA* SNPs and feed efficiency. Results: The duck *RXRA* gene was highly expressed on the fourth day of DPAC differentiation. The *RXRA* gene increased the content of fat and TG in DPACs and promoted the expression of cell differentiation genes; g.5,952,667 correlated with average daily feed intake (ADFI), residual feed intake (RFI), and feed conversion ratio (FCR). Conclusions: Duck *RXRA* can accelerate fat accumulation, and the polymorphism of the *RXRA* gene is closely related to feed efficiency, which provides basic data for breeding high feed efficiency ducks.

## 1. Introduction

In the production of meat ducks, feed cost accounts for 50–70% of the final cost [1]. Meat duck production is restricted by human competition for grains, and the feed cost gradually increases [2]. Residual feed intake (RFI) and feed conversion rate (FCR) are important indicators for measuring the growth traits of poultry. RFI is independent of growth performance and does not affect other important economic traits. It was proposed by Koch et al., and Luiting first used the term in poultry [3,4]. RFI is moderately inherited in poultry breeding. Research results from Bai et al. show that selecting a low RFI can improve the feed efficiency of poultry without affecting growth performance [1]. Therefore, it is necessary to improve feed efficiency in the growth of poultry. Fat metabolism plays a vital role in the change in feed efficiency [5,6,7,8]. The change in feed efficiency is related to lipid metabolism gene variations [9]. RFI-related genes were enriched in the lipid metabolism pathway in a whole genome sequencing study of high- and low-RFI groups of Beijing fatty chickens [10]. Moreover, numerous lipid metabolism genes were differentially expressed in Xiayan chickens with high- and low-RFIs [11].

Nuclear receptors (NRs) act as transcriptional regulators and control tissue development, homeostasis, and metabolism [12]. Retinoic acid X receptor alpha (*RXRA*) is a member of the nuclear receptor superfamily and participates in lipid metabolism, cell differentiation, and cell death [12,13,14]. *RXRA* expression is regulated by multiple factors. β-Hydroxybutyric acid further inhibits fatty acid oxidation and ketone body production by inhibiting the *RXRA* signal [15]. Long noncoding RNA (DANCR) combines with *RXRA* to enhance PI3K-AKT signal transduction and increase the occurrence of breast cancer [16]. The inhibition of the differentiation of preadipocytes is due to the inhibition of *RXRA* gene expression by miR-27a [17]. Moreover, *RXRA* as a transcription factor regulates downstream gene transcription. *RXRA* promotes the transcription of the perilipin 1 (PLIN1) gene to accelerate the differentiation of cells in the process of precursor fat differentiation, which occurs in a peroxisome proliferator-activated receptor gamma (*PPARG*)-independent manner [18].

At present, the effects of the *RXRA* gene on the adipogenesis and feed efficiency of ducks are not clear. Duck preadipocytes were cultured in vitro to observe the effect of the *RXRA* gene on lipid content. The relationship between growth traits and single nucleotide mutations of *RXRA* in the DNA of adult male ducks was analyzed, providing molecular markers for breeding duck growth traits.

## 2. Materials and Methods

### 2.1. Animal Experiments and Ethics Statement

The duck population (*n* = 500) in this study was provided by Qiangying Duck Breeding Co., Ltd. (Huangshan, China). The feeding and management methods of Jin et al. were referenced [7]. The daily food intake and weekly weight of the duck population were recorded during the experiment (21–42 d). The average daily feed intake (ADFI), average daily gain (ADG), RFI, and FCR of the ducks were calculated after the experiment. The animal experiments were conducted in accordance with the Regulations and Guidelines on Animal Management, and the experiments were approved by the Animal Care and Use Committee of Anhui Agricultural Capital Institution (no: SYXK 2016-007).

### 2.2. Cell Isolation, Culture, and Differentiation

In a sterile environment, 42-day-old male Cherry Valley ducks were selected and killed by venous bleeding. Duck preadipocytes were isolated according to the duck cell separation and culture method presented by Ding et al. [19]. During the cell culturing, the medium was replaced every two days.

Cells were reseeded into 96-well and 6-well plates for the experiment after 80% cell confluence. According to the instructions of the CCK-8 kit (Vazyme, Nanjing, China), the cell inoculation density was 2000 cells/well (*n* = 3), and the growth curve of cells was detected. During the 6-day process of cell differentiation, the differentiation medium was used for the culture, and the medium was changed and detected every two days. The differentiation medium consisted of a complete medium, 0.5 mmol/L of 3-isobutyl-1-methylxanthine (IBMX, Solarbio, Beijing, China), 1 μmol/L of dexamethasone (DEX, Solarbio, Beijing, China), 10 µg/µL of insulin (Sigma, Shanghai, China), and 300 µM of oleic acid (Sigma, Shanghai, China). The optimal induction time of duck preadipocytes was on the fourth day.

### 2.3. Plasmid Construction and Transfection

The complete CDS of *RXRA* (ENSAPLT00000013743.2, Ensembl) was cloned into the NheI and HindIII sites in the pBI-CMV3 vector (Pharma, Shanghai, China). The specific shRNA (5′-CCGGGGACAGGTCTTCAGGTAAACACTCGAGTGTTTACCTGAAGACCTGTCCTTTTTG-3′) of *RXRA* was designed and synthesized by Tsingke Biotechnology (Beijing, China). According to the manufacturer’s instructions, the vectors were transfected into cells using ExFect transfection reagent (Vazyme, Nanjing, China). The transfection conditions were ExFect: vector = 2:1.

### 2.4. Analysis of Triglyceride (TG) Content

Following the respective manufacturer’s instructions, a TG kit (Applygen, Beijing, China) was used to measure the TG content of cells, and a microplate reader (Bio-Tek, Winooski, VT, USA) was used to measure the TG content. In order to adjust different samples to the same level, a BCA kit (Vazyme, Nanjing, China) was used. According to the manufacturer’s instructions, the TG content was adjusted according to the amount of protein in the sample. For plasmid-transfected duck preadipocytes (DPACs), cells were collected 24 h after transfection.

### 2.5. Oil Red O-Staining

Oil red O stock solution: oil red O powder (0.2 g, Sigma, Shanghai, China) was dissolved in 40 mL of isopropanol overnight. Before staining, the stock solution was filtered with Xinhua™ No. 1 filter paper (Xinhua, Hangzhou, China). Working solution: The stock solution was diluted with water to a ratio of 6:4 and then filtered (Millipore, Shanghai, China) after standing for 10 min.

For fat staining and quantification, refer to Pan et al. [20].

### 2.6. RNA Isolation and Quantitative Real-Time PCR (RT-PCR)

Total RNA was extracted from cells with TRIzol (Thermo, Waltham, MA, USA), following the supplier’s protocol. Following the respective manufacturer’s instructions, 1 μg of RNA was reverse-transcribed into cDNA using an RT-cDNA synthesis kit (Vazyme, Nanjing, China). The RT-PCR (ABI7500, Waltham, MA, USA) was performed with a qPCR mix (Vazyme, Nanjing, China). The reaction system was 95 °C for 5 min, 95 °C for 30 s, and 60 °C for 30 s for 35 cycles. The mRNA level of the gene was detected by RT-PCR testing, and the *GAPDH* gene was used as a reference. The 2^−ΔΔCt^ method was used to calculate the relative expression of the expressed genes. Primer 5 software was used for the primer design. The primers were listed in Appendix A. The primer was synthesized by Shanghai Sangon Biotechnology Co., Ltd (Shanghai, China).

### 2.7. Western Blotting

After cell treatment, the cells were suspended in protein lysate (Meilunbio, Dalian, China), and then stood on ice for 10 min. We collected the supernatant centrifuged at 12,000 rpm (4 °C) for 10 min. A BCA kit (Vazyme, Nanjing, China) was used for the protein quantitative analysis [21]. The Western blotting experiments referred to the method of Jiang et al. [22]. The primary antibodies (RXRA, YN0018, GAPDH, YM3215, ImmunoWay, Plano, TX, USA) were diluted 1000 times by 1% BSA, and the secondary antibody (IgG, 30000-0-AP, Proteintech) was diluted 2000 times by 1% BSA. The protein was visualized using a highly sensitive ECL chemiluminescence detection kit (Vazyme, E412-01). ImageJ software (National Institutes of Health, Bethesda, MD, USA, v1.51) was used for the protein quantification.

### 2.8. PCR Amplification, Single Nucleotide Polymorphism (SNP) Detection, and Genotyping

The blood of 243 ducks were randomly selected from a population of 500 ducks. DNA was extracted from the whole blood of 243 ducks with a blood DNA extraction kit according to the manufacturer’s protocol (Tiangen, Beijing, China). The SNP primers (Forwards: 5′-GTCAGACCTGAGGGCACAA-3′; Reverse: 5′-GCTCACCGCAACCATACA-3′) for the *RXRA* gene were designed by Primer 5 software and synthesized by Shanghai Sangon Biotechnology Co., Ltd. The sequence of the duck *RXRA* gene (g.5,953,498~g.5,952,805) was amplified by PCR in a 20 μL reaction volume, including 10 μL of Taq PCR Master Mix (Vazyme, Nanjing, China), 0.4 μL of forwards and reverse primers (10 μM), 1.0 μL of DNA, and 8.2 μL of nuclease-free H_2_O. The PCR conditions were as follows: 95 °C for 3 min; 30 cycles of 95 °C for 30 s, 55.2 °C for 30 s, and 72 °C for 5 s; and a final 5 min extension at 72 °C. PCR products were detected by 1% agarose gel electrophoresis and sent to Sangon for sequencing. The target peaks in the sequencing chromatographs obtained for ducks were labelled and analyzed using DNA star software (DNASTAR, Madison, WI, USA, v11.1) to screen for mutation sites. One individual was not completely sequenced, so there were 6 loci that could not be counted.

### 2.9. Polymorphisms in RXRA and Their Associations with Growth Traits

SPSS 25 software was used to establish a linear model of SNPs (haplotypes) and traits and correlation analysis.

The mixed linear model that was used is as follows:Y = u + G + B + e

Y: feed efficiency; u: overall population mean; G: fixed effect of genotype or haplotype; B: birth weight; and e: random residual error.

### 2.10. Statistical Analysis

The cell level results in this study are presented as the mean ± SD, and the difference between the groups was analyzed by a one-way ANOVA test. Pop-gene software (a joint Project of the Molecular Biology and Biotechnology Centre, University of Alberta and the Center for International Forestry Research, v1.32, Edmonton, Alberta, Canada) was used to calculate gene frequency, genotype frequency, genetic heterozygosity, effective allele number, and polymorphism information content. Haploview software (Daly Lab at the Broad Institute, v4.2, Cambridge, Cambs, UK) was used to analyze the haplotype relationship of mutation sites. The population-level results in this study are presented as the mean ± SEM. The association analysis of duck growth traits was performed using the Pearson product-moment correlation coefficient. The LSD test was used to analyze the significance of the associations between traits and SNPs, and *p* < 0.05 was considered a significant correlation. The different letters (^a,b,c^) represent significant differences.

## 3. Results

### 3.1. Fat Accumulation and RXRA Expression Pattern in DPACs

To investigate the expression of the duck *RXRA* gene during the differentiation of DPACs, duck abdominal preadipocytes were isolated. The generality of DPACs was explored in this study, and the results showed “S” curve proliferation at 0–6 days (Figure 1A). In the process of cell differentiation, the intracellular fat content gradually increased every two days (Figure 1B,C). The TG content on days 4 and 6 of cell differentiation was higher than that on days 0 and 2 (Figure 1D). The expression of *PPARG* and *C/EBPA* genes on the day 6 was higher than that on days 0, 2, and 4 (Figure 1E,F). The expression of *RXRA* mRNA and protein was the highest on the fourth day of cell differentiation (Figure 1G, H, I). This evidence drew our attention to the potential role of *RXRA* in duck fat metabolism.

### 3.2. Effect of Duck RXRA on Fat Metabolism of DPACs

The DPACs were transfected with OERXRA, shRNA, and normal control vector, and a large amount of green fluorescence occurred in the cells; however, no fluorescence occurred in the cells supplemented with the culture medium (Figure 2A). The RT-PCR results showed that the expression of *RXRA* after overexpression was significantly higher than that of the normal control group, whereas the expression of *RXRA* after knockdown was significantly lower than that of the control group (Figure 2B). *RXRA* protein expression was consistent with mRNA expression (Figure 2C). The OERXRA group increased the content of lipid droplets in cells compared with the normal and negative control groups, and the knockdown of *RXRA* inhibited the content of lipid droplets in cells compared with the normal and negative control groups (Figure 2D,E). Compared with the control group, OERXRA increased TG content, whereas shRXRA decreased TG content (Figure 2F). OERXRA promoted the mRNA expression of *RXRA* and *C/EBPA* genes; in contrast, shRXRA inhibited the mRNA expression of *RXRA* and *C/EBPA* genes in cells (Figure 2G,H).

### 3.3. Single Nucleotide Polymorphism Site (SNPs) and Genotyping of the RXRA Gene

According to the Ensembl sequencing results (Ensembl ID: ENSAPLT00000013743.2) and PCR sequencing map, four SNPs (g.5,952,703 C > T, g.5,952,667 C > T, g.5,952,666 G > A, and g.5,952,658 C >T (Figure 3A–D)) were identified in the coding region of the *RXRA* gene and eight SNPs (g.5,952,604 C > T, g.5,952,543 G > A, g.5,952,400 C > T, g.5,952,353 G > A, g.5,952,343 C > T, g.5,952,201 T > C, g.5,952,197 C > T, and g.5,952,188 A > T (Figure 3E–L)) were identified in the noncoding region of the duck population. The 12 SNPs were counted and calculated for polymorphism, as shown in Table 1. The allele found in the database is defined as wild-type, and the mutant allele shown by the sequencing results is defined as mutant-type. In the seven SNPs of g.5,952,666 G > A, g.5,952,658 C >T, g.5,952,604 C > T, g.5,952,543 G > A, g.5,952,353 G > A, g.5,952,343 C > T, and g.5,952,197 C > T, the frequency of wild-type alleles was greater than the frequency of mutant alleles. In addition, in the five SNPs of g.5,952,703 C > T, g.5,952,667 C > T, g.5,952,400 C > T, g.5,952,201 T > C, and g.5,952,188 A > T, the frequency of wild-type alleles is less than mutant alleles. Genetic diversity is very important for the genetic potential of a species, and it is also the basis for breeding excellent varieties. Polymorphic information content (PIC) refers to the value of a marker that is used to detect polymorphisms in a population. According to the classification of PIC values (PIC < 0.25, low polymorphism; 0.25 < PIC < 0.5, medium polymorphism; and PIC > 0.5, high polymorphism), the 12 SNPs in this population ranged from 0.30 to 0.32, belonging to the medium polymorphism population. According to the Hardy–Weinberg principle, the 12 SNPs of this population reached a genetic balance of *p* > 0.05 (Table 1).

### 3.4. Analysis of Amino Acids in the Coding Region of the RXRA Gene

Through sequence alignment, single nucleotide mutations were found to cause amino acid changes in the duck population, where g.5,952,703 C > T causes the missense mutation of proline at position 28 to leucine (Figure 4A) and g.5,952,667 C > T causes an alanine missense mutation at position 40 to valine (Figure 4B); g.5,952,666 G > A and g.5,952,658 C > T are synonymous mutations (Figure 4C,D).

### 3.5. Correlation Analysis of Duck Feed Efficiency Traits

The growth and feed efficiency traits were measured from the *n* = 243 population that were used in the genotype association analysis. To study the correlation among initial body weight (IBW, 21 d, 1456.90 ± 5.80 g), final body weight (FBW, 42 d, 4000.70 ± 19.62 g), total feed intake (FI, 4636.57 ± 29.48 g), ADFI, ADG, RFI, and FCR, Pearson’s correlation coefficient was calculated. As shown in Table 2, the IBW was correlated with the FBW, FI, ADFI, ADG, RFI, and FCR (*p* < 0.05). The FBW was correlated with the FI, ADFI, ADG, and FCR (*p* < 0.05). The FI was correlated with the ADFI, ADG, RFI, and FCR (*p* < 0.05). The ADFI was correlated with the ADG, RFI, and FCR (*p* < 0.05). The ADG was negatively correlated with the FCR (*p* < 0.01). The RFI correlated with the FCR (*p* < 0.01).

A relationship between SNPs and growth traits was discovered following the mixed linear model analysis. The SNPs that were significantly associated with a trait or that differed between phenotypes are listed in Table 3. All SNPs were significantly related to the RFI. g.5,952,667 C > T, g.5,952,666 G > A, g.5,952,658 C >T, and g.5,952,201 T > C were significantly related to the FCR. Except for g.5,952,604 G > A, g.5,952,543 G > A, g.5,952,353 G > A, g.5,952,343 C > T, and g.5,952,197 C > T, ADFI was related with the remaining 7 SNPs, with all being at significant levels. None of the SNPs were associated with the IBW, FBW, FI, and ADG. Favorite among all SNPs, the TT genotype of g.5,952,667 C > T had the lowest ADFI, RFI, and FCR, whereas the CC genotype had the highest ADFI. The highest RFI belonged to the AA genotype of g.5,952,666 G > A. The CC genotype of g.5,952,658 C >T and the AA genotype with g.5,952,666 G > A had the highest FCR.

### 3.6. Linkage Disequilibrium Analysis and Haplotype Analysis of SNPs of the RXRA Gene

The results of the linkage disequilibrium analysis for the *RXRA* SNPs are shown in Figure 5. The LD map shows the SNPs of the *RXRA* gene, and vital LD regions were detected. There are two haploids in this block. The frequencies of H1 (TCCCGTGCCGTT) and H2 (ATTTACATTACC) were 0.717 and 0.240, respectively. The relationship between haplotype and growth traits was analyzed. The analysis results are shown in Table 4. The ADFI, RFI, and FCR of the H1H1 haplotype combination were lower than those parameters of the H1H2 and H2H2 groups and had no correlation with the ADG.

## 4. Discussion

*RXRA*, a member of the nuclear hormone receptor superfamily, is involved in regulating lipid, glucose, and energy metabolism [23,24,25]. Fat metabolism is an important factor affecting the growth traits of meat ducks. Therefore, we explored the role of *RXRA* lipid metabolism in vitro. In the process of inducing poultry preadipocytes to mature adipocytes in vitro, oleic acid has excellent differentiation efficiency [19,26,27]. The duck preadipocytes isolated in this study showed similar results; the optimal induction time of duck preadipocytes was the fourth day. *RXRA* and *C/EBPA* genes have been reported to be crucial in adipocyte differentiation [19,28,29]. Interestingly, the expression pattern of *RXRA* was the same as the expression pattern of *RXRA* and *C/EBPA* in the first four days of cell lipid accumulation. At present, *RXRA* may play a key role in duck fat accumulation.

*RXRA* can bind PPAR response elements of *RXRA* to form dimers, which activate the transcription of downstream genes to regulate lipid metabolism [30]. The expression of *C/EBPA* was regulated by *RXRA* and *RXRA* polymer; then, *C/EBPA* combined with *RXRA* stimulated adipocyte differentiation [31]. In this study, *RXRA* promoted the expression of *RXRA* and *C/EBPA* and promoted the accumulation of intracellular lipids. Noteworthily, *RXRA* may promote intracellular fat accumulation in duck preadipocytes through the *RXRA*-*C/EBPA* pathway. *RXRA* can regulate cell differentiation by binding cytokines, and it also has important functions on lipid metabolism. In obese mouse models, RXR agonists can reduce food intake and weight gain to maintain the balance of blood glucose and insulin sensitivity [32]. However, feeding *RXRA* knockout mice a high-fat diet can cause dyslipidemia [33]. The study of the influence of *RXRA* on TG is difficult to explain. LG268 is an agonist of *RXRA*, which can reduce TG in plasma [34]. When an *RXRA* agonist was fed to different animals, the serum triglyceride content changed [35,36].

The duck *RXRA* gene is located on chromosome 18 between 6.20 and 6.34 Mb [37]. The *RXRA* gene of cattle is located in 105.98–106.01 Mb of chromosome 1, and its QTLs are related to growth performance and meat quality [23]. The *RXRA* gene of pigs is located at 288.83–288.86 Mb on chromosome 1, and its QTLs are correlated with abdominal fat, back fat, body weight, daily gain, and meat quality [38,39,40]. The *RXRA* gene of chickens is located at 7.27–7.35 Mb on chromosome 17, and its QTLs are related to abdominal fat and body weight [41]. The function and location of *RXRA* are not standard, but its function is similar to that of mammals and chickens and is related to fat metabolism. *RXRA* can be used as a candidate gene related to meat duck production and related to production traits. As an effective means of measuring the feed efficiency of ducks, the residual feed intake usually changes during the individual life cycle, and the heritable components of the variation may also show individual genetic variation.

In this study, there were 12 mutation sites in the *RXRA* gene sequence in the duck population (*n* = 243), and the PIC belonged to moderate polymorphism. In addition, the SNPs are in Hardy–Weinberg equilibrium and can be used as genetic markers in the population. Surprisingly, g.5,952,703 C > T changed the 28th amino acid (proline) of *RXRA* to leucine. The mutation of the base in the coding region may lead to abnormal transcription, translation, and gene function changes [42]. Although the SNPs in the intron could not cause amino acid sequence changes, synonymous mutations may regulate gene expression due to affecting gene transcription by changing the DNA sequence of transcriptional regulatory sites [43]. Proline has excellent performance in maintaining blood glucose levels and improving tolerance [44]. Leucine can inhibit food intake and fat transport [45]. Feed intake is well-known to be a key factor of feed efficiency. When proline is changed to leucine, leucine may reduce the feed weight ratio of ducks by inhibiting feed intake and balancing the fat energy supply. In addition, g.5,952,667 C > T causes the 40th amino acid (alanine) of the *RXRA* gene sequence to change to valine. Ala improves glucose metabolism to increase body energy [23]. Valine can promote PPARA-dependent fatty acid oxidation and the de novo synthesis of Pr-CoA fatty acids [45]. In this study, homozygous valine individuals (TT) had the lowest RFI and FCR values. This result may be due to *RXRA* participating in regulating the fat metabolism of meat ducks and its specific mechanism still being unclear. Although the amino acids of g.5,952,666 G > A and g.5,952,658 C > T did not change, the transcription speed might change, leading to changes in growth traits. SNPs in noncoding regions also significantly affect transcriptional efficiency, gene expression, and splicing dysregulation [46,47]. Noteworthily, these 12 mutation sites have a strong linkage relationship. The haplotype of H1H1 account for half of the population, and their feed efficiency is more popular.

## 5. Conclusions

In conclusion, the duck *RXRA* gene facilitates preadipocyte fat accumulation and preadipocyte differentiation via the *RXRA*-*C/EBPA* signal. In addition, the SNP of *RXRA* is correlated with the duck ADFI, RFI, and FCR, therefore suggesting a potential molecular marker of *RXRA* in duck feed efficiency traits.

## Figures and Tables

**Figure 1 animals-13-00680-f001:**
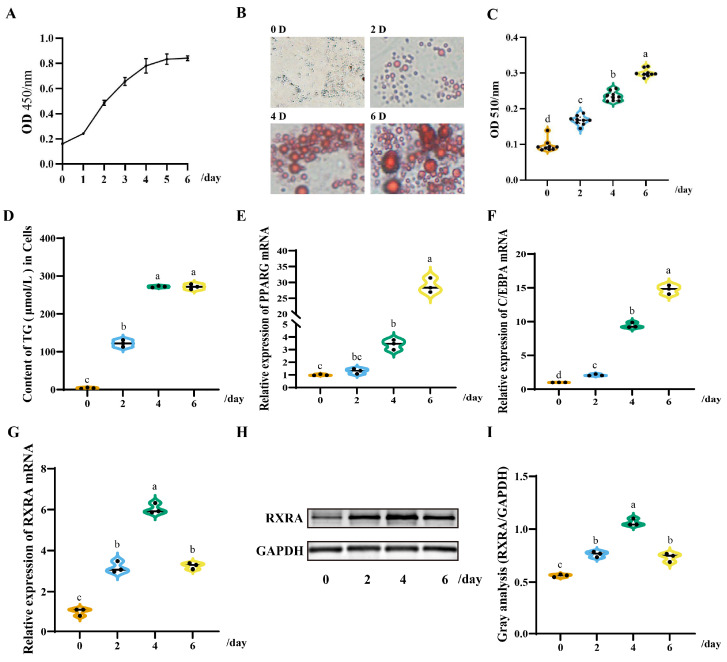
The expression pattern of duck *RXRA* in the duck preadipocytes (DPACs). (**A**) Multiplication curve of the DPACs. (**B**) Fat content of the DPACs induced by the differentiation medium at 0, 2, 4, and 6 days. The images were taken under 200× magnification. (**C**) OD 510/nm: absorbance at 510 nm. (**D**) Triglyceride (TG) content at 0, 2, 4, and 6 days of the induced DPACs. (**E**) The genes expression changes of the induced DAPCs. (**E**) The *PPARG* gene expression of the induced DAPCs. (**F**) The *C/EBPA* gene expression of the induced DAPCs. (**G**) The *RXRA* gene expression of the induced DAPCs. (**H,I**) Changes in RXRA protein expression in the DPACs induced. The data are reported as the mean ± SD on the basis of *n* = 3 independent experiments. ^a,b,c,d^ Significant differences are shown in superscript letters, with different letters representing significant differences (*p* < 0.05). Original Western blot figures in Appendix A.

**Figure 2 animals-13-00680-f002:**
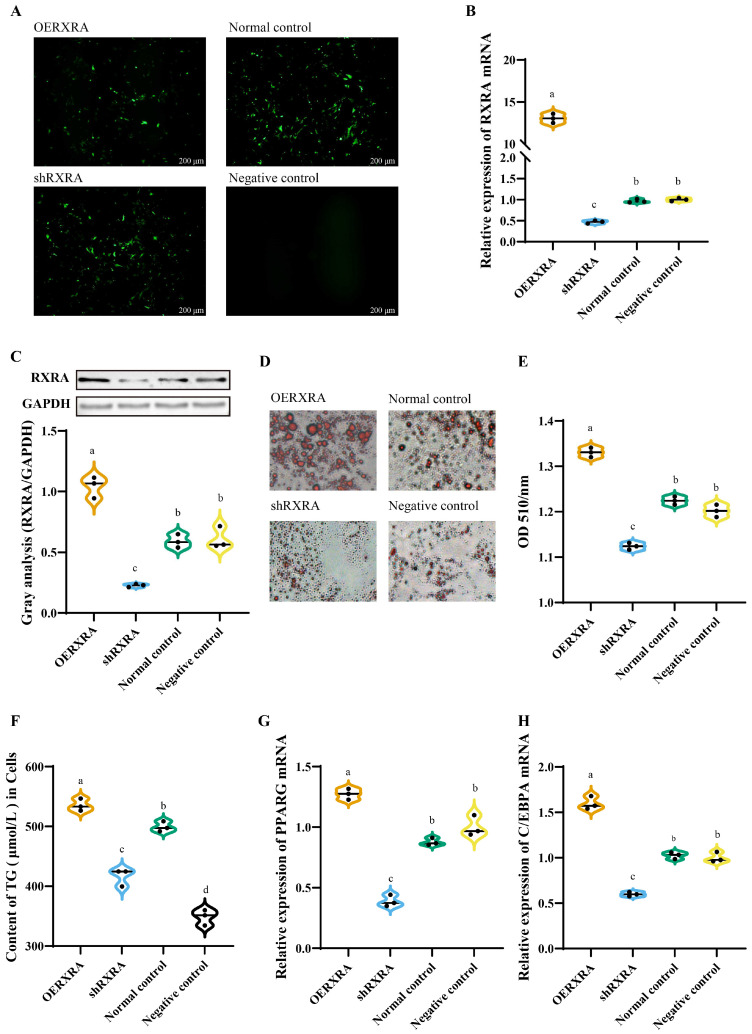
Effect of *RXRA* gene on the lipid metabolism of the DPACs. (**A**) The transfections of overexpression *RXRA* (OERXRA), knockdown *RXRA* (shRXRA), blank plasmid (Normal control), and culture medium (negative control) into the DPACs were observed under a fluorescence microscope. (**B**) The *RXRA* gene mRNA expression. (**C**) The indicated protein levels were detected by a Western blot analysis, and the relative fold change in RXRA/GAPDH determined by Western blotting was quantified through a grayscale scan. (**D**) Intracellular fat content. The images were taken under 200× magnification. (**E**) OD 510/nm: absorbance at 510 nm. (**F**) TG content in the DPACs. (**G**) The mRNA expression of *PPARG* gene. (**H**) The mRNA expression of *C/EBPA* gene. The data are reported as the mean ± SD on the basis of *n* = 3 independent experiments. ^a,b,c,d^ Significant differences are shown in superscript letters, with different letters representing significant differences (*p* < 0.05). Original Western blot figures in Appendix A.

**Figure 3 animals-13-00680-f003:**
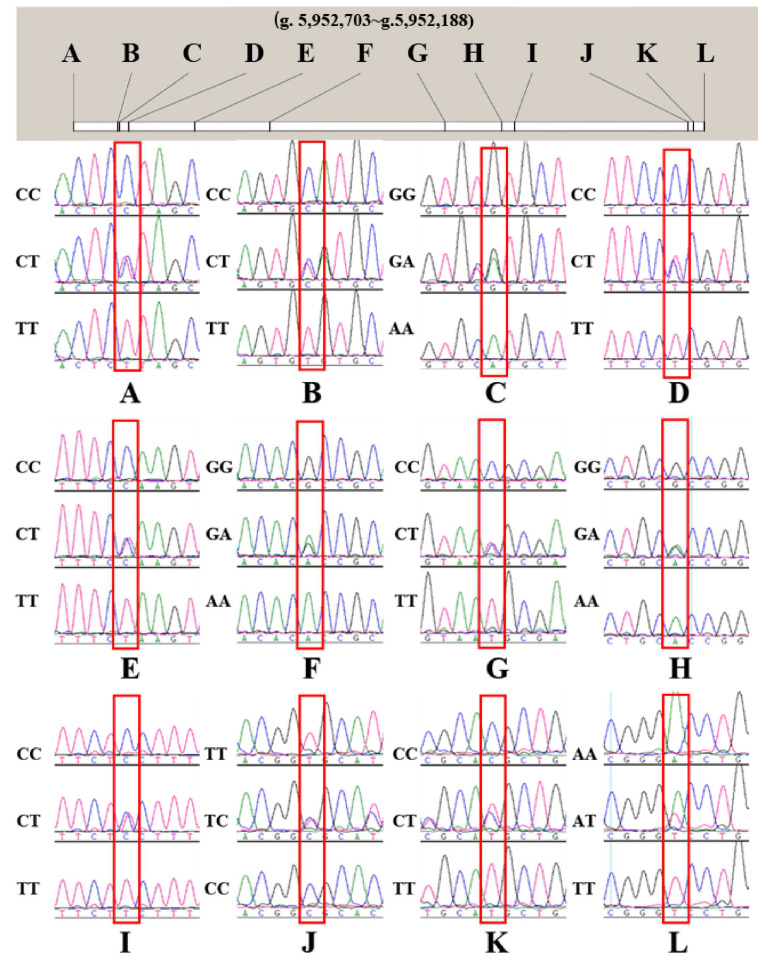
The peak maps of all genotypes of the SNPs. (**A**): g.5,952,703 C > T; (**B**) g.5,952,667 C > T. (**C**) g.5,952,666 G > A. (**D**) g.5,952,658 C >T. (**E**) g.5,952,604 C > T. (**F**) g.5,952,543 G > A. (**G**) g.5,952,400 C > T. (**H**) g.5,952,353 G > A. (**I**) g.5,952,343 C > T. (**J**) g.5,952,201 T > C. (**K**) g.5,952,197 C > T. (**L**) g.5,952,188 A > T.

**Figure 4 animals-13-00680-f004:**
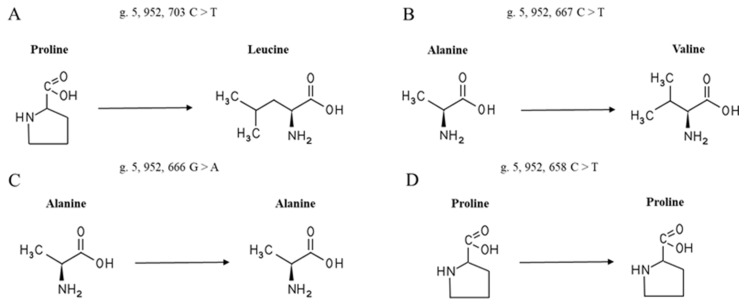
The change in the *RXRA* amino acid.

**Figure 5 animals-13-00680-f005:**
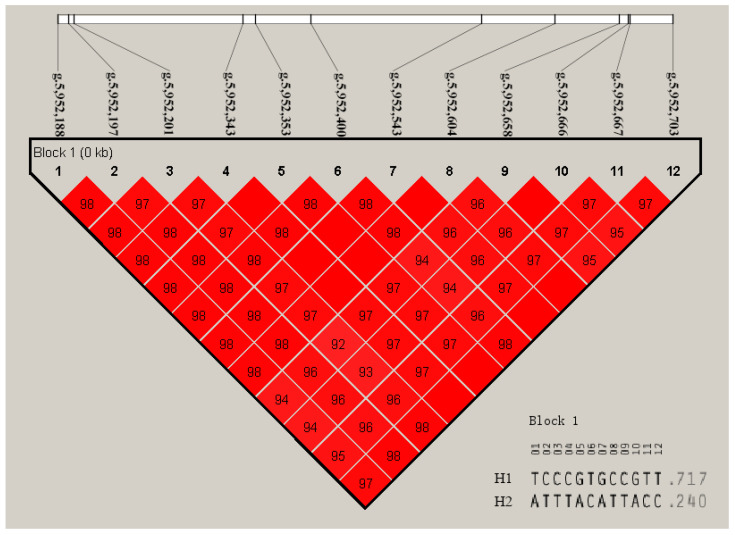
The linkage disequilibrium analysis for the *RXRA* SNPs.

**Table 1 animals-13-00680-t001:** Single nucleotide genetic polymorphism analysis of the *RXRA* gene.

SNPs	Genotype Frequency	Allele Frequency	PIC	χ^2^	*p*
g.5,952,703 C > T	TT (0.5391) TC (0.3868) CC (0.0741)	T (0.7325) C (0.2675)	0.3151	0.0544	0.8156
g.5,952,667 C > T	TT (0.5391) TC (0.3868) CC (0.0741)	T (0.7325) C (0.2675)	0.3151	0.0544	0.8156
g.5,952,666 G > A	AA (0.0700) AG (0.3703) GG (0.5597)	A (0.2551) G (0.7449)	0.3078	0.1861	0.6662
g.5,952,658 C >T	TT (0.0700) TC (0.3745) CC (0.5555)	T (0.2572) C (0.7428)	0.3091	0.1861	0.6662
g.5,952,604 C > T	TT (0.0741) TC (0.3662) CC (0.5597)	T (0.2572) C (0.7428)	0.3091	0.4615	0.4969
g.5,952,543 G > A	AA (0.0741) AG (0.3662) GG (0.5597)	A (0.2572) G (0.7428)	0.3091	0.4615	0.4969
g.5,952,400 C > T	TT (0.5432) TC (0.3817) CC (0.0741)	T (0.7346) C (0.2654)	0.3139	0.1036	0.7475
g.5,952,353 G > A	AA (0.0700) AG (0.3703) GG (0.5597)	A (0.2551) G (0.7449)	0.3078	0.1861	0.6662
g.5,952,343 C > T	TT (0.0700) TC (0.3703) CC (0.5597)	T (0.2551) C (0.7449)	0.3078	0.1861	0.6662
g.5,952,201 T > C	CC (0.5473) CT (0.3827) TT (0.0700)	C (0.7387) T (0.2613)	0.3115	0.028	0.8671
g.5,952,197 C > T	TT (0.0700) TC (0.3703) CC (0.5597)	T (0.2551) C (0.7449)	0.3078	0.1861	0.6662
g.5,952,188 A > T	AA (0.0782) AT (0.3827) TT (0.5391)	T (0.7305) A (0.2695)	0.3162	0.2216	0.6378

**Table 2 animals-13-00680-t002:** Correlation coefficients (r) between feed efficiency.

	IBW	FBW	FI	ADFI	ADG	RFI	FCR
IBW	1	0.463 **	0.296 **	0.296 **	0.185 **	0.198 **	0.135 *
FBW	0.463 **	1	0.795 **	0.795 **	0.957 **	−0.026	−0.397 **
FI	0.296 **	0.795 **	1	1.000 **	0.785 **	0.536 **	0.150 *
ADFI	0.296 **	0.795 **	1.000 **	1	0.785 **	0.536 **	0.150 *
ADG	0.185 **	0.957 **	0.785 **	0.785 **	1	−0.094	−0.484 **
RFI	0.198 **	−0.026	0.536 **	0.536 **	−0.094	1	0.866 **
FCR	0.135 **	−0.397 **	0.150 *	0.150 *	−0.484 **	0.866 **	1

Note: IBW: initial body weight, FBW: final body weight, ADFI: average daily feed intake, ADG: average daily gain, RFI: residual feed intake, FCR: feed conversion rate. * Indicates a significant correlation at the level of 0.05 (two-tailed), and ** indicates a significant correlation at the level of 0.01 (two-tailed).

**Table 3 animals-13-00680-t003:** Association analysis of *RXRA* SNPs with feed efficiency traits (mean ± SEM).

SNPs	Genotype Frequency/*n*	IBW	FBW	FI	ADFI	ADG	RFI	FCR
g.5,952,703 C > T	TT (131)	1455.41 ± 8.27	3985.26 ± 25.01	4594.92 ± 37.83	217.74 ± 1.91 ^b^	120.62 ± 1.16	−1.96 ± 1.24 ^c^	1.81 ± 0.01
	CT (94)	1454.23 ± 8.84	4033.00 ± 31.46	4692.52 ± 47.25	223.96 ± 2.19 ^a^	122.49 ± 1.27	1.67 ± 1.24 ^b^	1.83 ± 0.01
	CC (18)	1484.08 ± 20.1	3951.31 ± 104.34	4647.50 ± 147.64	226.40 ± 5.53 ^a^	117.83 ± 3.79	10.20 ± 3.74 ^a^	1.94 ± 0.05
g.5,952,667 C > T	TT (131)	1455.93 ± 8.29	3985.37 ± 25.02	4590.36 ± 38.33	217.24 ± 1.92 ^b^	120.66 ± 1.16	−2.53 ± 1.23 ^c^	1.81 ± 0.01 ^b^
	CT (94)	1454.24 ± 8.84	4033.01 ±31.46	4692.52 ± 47.25	223.96 ± 2.19 ^a^	122.49 ± 1.27	1.67 ± 1.24 ^b^	1.83 ± 0.01 ^b^
	CC (18)	1480.31 ± 20.1	3950.53 ± 104.32	1680.67 ± 139.33	230.10 ± 4.85 ^a^	117.52 ± 3.80	14.33 ± 2.93 ^a^	1.98 ± 0.05 ^a^
g.5,952,666 G > A	AA (18)	1483.5 ± 21.07	3965.74 ± 109.47	4700.37 ± 146.29	231.47 ± 4.93 ^a^	118.09 ± 3.98	14.81 ± 3.07 ^a^	1.98 ± 0.05 ^a^
	AG (89)	1455.78 ± 8.94	4025.10 ± 32.76	4683.27 ± 48.78	223.54 ± 2.26 ^a^	122.03 ± 1.31	1.91 ± 1.27 ^b^	1.84 ± 0.01 ^b^
	GG (136)	1454.65 ± 8.13	3989.79 ± 24.34	4597.69 ± 37.44	217.63 ± 1.87 ^b^	120.92 ± 1.13	−2.50 ± 1.20 ^c^	1.81 ± 0.01 ^b^
g.5,952,658 C >T	CC (18)	1483.5 ± 21.07	3965.74 ± 107.47	4700.38 ± 146.29	229.99 ± 4.88 ^a^	117.91 ± 3.76	13.69 ± 3.11 ^a^	1.97 ± 0.05 ^a^
	CT (91)	1454.44 ± 8.94	4024.36 ± 32.40	4684.12 ± 48.25	223.57 ± 2.23 ^a^	122.06 ± 1.30	1.90 ± 1.25 ^b^	1.84 ± 0.01 ^b^
	TT (134)	1455.54 ± 8.14	3990.02 ± 24.52	4596.49 ± 37.70	217.66 ± 1.90 ^b^	120.94 ± 1.14	−2.50 ± 1.21 ^c^	1.81 ± 0.01 ^c^
g.5,952,604 C > T	TT (18)	1484.08 ± 20.10	3951.31 ± 104.34	4678.50 ± 147.64	226.4 ± 5.53	117.83 ± 3.79	10.20 ± 3.74 ^a^	1.94 ± 0.05
	CT (89)	1456.12 ± 9.04	4028.88 ± 32.91	4687.06 ± 49.18	223.73 ± 2.28	122.19 ± 1.32	1.86 ± 1.28 ^b^	1.84 ± 0.01
	CC (136)	1454.15 ± 8.12	3989.68 ± 24.34	4602.57 ± 29.48	218.12 ± 1.86	120.89 ± 1.13	−1.95 ± 1.21 ^c^	1.81 ± 0.01
g.5,952,543 G > A	AA (18)	1484.08 ± 20.10	3951.31 ± 104.34	4647.50 ± 147.64	226.4 ± 5.53	117.83 ± 3.79	10.20 ± 3.74 ^a^	1.94 ± 0.05
	AG (89)	1456.12 ± 9.04	4028.88 ± 32.91	4687.06 ± 49.18	223.73 ± 2.28	122.19 ± 1.32	1.86 ± 1.28 ^b^	1.84 ± 0.01
	GG (136)	1454.15 ± 8.12	3989.69 ± 24.34	4602.08 ± 36.96	218.12 ± 1.86	120.89 ± 1.13	−1.95 ± 1.21 ^c^	1.81 ± 0.01
g.5,952,400 C > T	TT (132)	1456.79 ± 8.22	3990.55 ± 25.01	4595.69 ± 37.64	217.79 ± 1.90 ^b^	120.80 ± 1.16	−2.19 ± 1.22 ^c^	1.81 ± 0.01
	CT (92)	1452.25 ± 8.92	4025.94 ± 31.62	4695.66 ± 48.04	224.12 ± 2.22 ^a^	122.28 ± 1.29	2.13 ± 1.27 ^b^	1.94 ± 0.01
	CC (18)	1484.08 ± 20.10	3951.31 ± 104.34	4647.50 ± 147.64	226.40 ± 5.53 ^a^	117.83 ± 3.79	10.20 ± 3.74 ^a^	1.94 ± 0.05
g.5,952,353 G > A	AA (17)	1487.50 ± 21.01	3966.56 ± 109.48	4665.26 ± 155.46	227.54 ± 5.73	118.41 ± 3.97	10.44 ± 3.96 ^a^	1.94 ± 0.05
	AG (89)	1455.78 ± 8.94	4025.10 ± 32.76	4686.46 ± 49.22	223.70 ± 2.28	122.06 ± 1.33	2.00 ± 1.28 ^b^	1.84 ± 0.01
	GG (136)	1454.15 ± 8.12	3989.69 ± 24.34	4602.07 ± 36.96	218.12 ± 1.86	120.88 ± 1.12	−1.95 ± 1.21 ^c^	1.81 ± 0.01
g.5,952,343 C > T	TT (17)	1487.5 ± 21.01	3966.56 ± 109.48	4665.26 ± 155.46	227.54 ± 5.73	118.41 ± 3.97	10.44 ± 3.96 ^a^	1.94 ± 0.05
	CT (89)	1455.78 ± 8.94	4025.10 ± 32.76	4686.46 ± 464.38	223.70 ± 2.28	122.06 ± 1.33	2.00 ± 1.28 ^b^	1.84 ± 0.01
	CC (136)	1454.15 ± 8.12	3989.69 ± 24.34	4602.08 ± 36.96	218.12 ± 1.86	120.88 ± 1.12	−1.95 ± 1.21 ^c^	1.81 ± 0.01
g.5,952,201 T > C	CC (133)	1487.06 ± 21.09	3937.88 ± 109.75	4597.42 ± 37.37	217.89 ± 1.88 ^b^	120.63 ± 3.75	−1.87 ± 1.24 ^b^	1.81 ± 0.01 ^b^
	CT (92)	1454.00 ± 8.83	4026.83 ± 31.61	4692.78 ± 48.09	223.98 ± 2.22 ^a^	122.24 ± 1.29	2.06 ± 1.27 ^a^	1.84 ± 0.01 ^b^
	TT (17)	1455.40 ± 8.21	3991.35 ± 24.86	4652.59 ± 156.51	226.84 ± 5.84 ^a^	119.19 ± 1.88	8.80 ± 3.68 ^a^	1.92 ± 0.05 ^a^
g.5,952,197 C > T	TT (17)	1487.50 ± 21.01	3966.56 ± 109.48	4665.26 ± 155.46	227.54 ± 5.73	118.41 ± 3.97	10.44 ± 3.96 ^a^	1.94 ± 0.05
	CT (89)	1454.34 ± 8.94	4016.68 ± 32.35	4683.16 ± 49.05	223.70 ± 2.28	122.06 ± 1.33	2.00 ± 1.28 ^b^	1.84 ± 0.01
	CC (136)	1455.10 ± 8.12	3995.19 ± 24.62	4604.24 ± 37.10	218.12 ± 1.86	120.88 ± 1.12	−1.95 ± 1.21 ^c^	1.81 ± 0.01
g.5,952,188 A > T	TT (131)	1457.05 ± 8.28	3991.35 ± 25.21	4598.86 ± 37.92	217.93 ± 5.28 ^b^	120.83 ± 1.17	−2.10 ± 1.23 ^c^	1.81 ± 0.01
	AT (92)	1452.27 ± 8.92	4025.30 ± 31.61	4692.76 ± 48.09	223.98 ± 2.23 ^a^	122.25 ± 1.29	2.04 ± 1.27 ^b^	1.84 ± 0.01
	AA (19)	1480.71 ± 19.31	3950.95 ± 98.70	4636.92 ± 140.06	225.63 ± 5.28 ^a^	117.95 ± 3.58	9.33 ± 3.65 ^a^	1.93 ± 0.05

Note: IBW: initial body weight, FBW: final body weight, ADFI: average daily feed intake, ADG: average daily gain, RFI: residual feed intake, FCR: feed conversion rate. ^a,b,c^ Significant differences are shown in superscript letters, with different letters representing significant differences (*p* < 0.05).

**Table 4 animals-13-00680-t004:** Association analysis between haplotype and feed efficiency traits.

Traits	Haplotype (Mean ± SE)		
	H1H1 (128)	H1H2 (85)	H2H2 (16)
IBW	1456.25 ± 8.45	1454.45 ± 9.30	1490.87 ± 22.08
FBW	3987.18 ± 25.56	4021.97 ± 33.54	3953.25 ± 115.69
FI	4596.63 ± 38.63	4687.40 ± 51.15	4671.78 ± 165.36
ADFI	217.50 ± 1.94 ^b^	223.19 ± 2.34 ^a^	228.09 ± 6.08 ^a^
ADG	120.83 ± 1.19	121.97 ± 1.36	119.89 ± 3.92
RFI	−2.40 ± 1.22 ^b^	2.22 ± 1.32 ^a^	8.97 ± 3.92 ^a^
FCR	1.81 ±0.01 ^b^	1.83 ± 0.01 ^b^	1.92 ± 0.05 ^a^

Note: ^a,b^ Significant differences are shown in superscript letters, with different letters representing significant differences (*p* < 0.05).

## Data Availability

The data related to this paper may be requested from the corresponding author.

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
