# Peer review of "The Duck RXRA Gene Promotes Adipogenesis and Correlates with Feed Efficiency"

_animals, 2023, doi:10.3390/ani13040680_

Round 1

Reviewer 1 Report

You can find my comments in the pdf file.

Author Response

Thank you for your valuable comments.

Our response can be found in the attachment.

Reviewer 2 Report

In this manuscript, the authors showed that the duck RXRA gene is involved in adipogenesis using preadipocytes overexpressing or knockdown the RXRA gene. Moreover, they reported that SNPs within this gene are associated with feed efficiency traits by studying more than 200 ducks.

Improving feed efficiency in livestock animals is a major goal for the industry, and the involvement of the RXRA gene in feed efficiency and lipid formation has not been previously reported in ducks. Their work has novelty and is in line with the journal's aim. However, there are some points that need to be addressed before publication.

Major comments:
1. The authors should write the abstract following the journal guideline. According to the journal guideline, the abstract should be about 200 words maximum. The abstract should follow the style of structured abstracts but without headings. The author's abstract contains more than 300 words and has headings.
2. The author suggested a possibility of RXRA as a molecular marker for duck breeding in feed efficiency traits in the conclusion (Lines 32-34 and 380-383), however, it was not discussed in the contents. Please describe how the obtained results of RXRA may be used for duck breeding.
3. The authors should provide information on the Cherry Vally ducks' body weight. Also, the SNP analysis should analyze the effect on final body weight in addition to feed efficiency.
4. It is strange that there is no citation in the Materials and Methods section. Moreover, some parts are very similar to those in the authors' recently accepted papers (Pan et al. (2023). A Novel in Duck Myoblasts: The Transcription Factor Retinoid X Receptor Alpha (RXRA) Inhibits Lipid Accumulation by Promoting CD36 Expression. International Journal of Molecular Sciences, 24, 1180.). Please use references appropriately and avoid possible self-plagiarism.
5. The authors did not discuss their results obtained from the experiment using the cells in the Discussion section. Most information in the Discussion section should be written in the Introduction section to explain their research background and objectives in detail.

Minor comments:
Line 43: The name of the author must be accurate. 'Lui' should be 'Luiting'.
Line 51: Please change XIAYAN to Xiayan.
Lines 55-58: What does the author want to tell using the two references 15 and 16? These references do not describe the previous sentence that RXRA expression is regulated by multiple factors.
Lines 61-64: Why does the author not examine the PLIN1 gene in this study?
Lines 83-85: Please show the body weight information. How similar is it?
Lines 103-105: Is this operation the same as the previous sentence (Lines 101-103)?
Lines 129-130: What are the CS2 cells? Where do the overexpression and the interference plasmids come from?
Lines 149-151: Did the author use 2 different methods to extract total RNA? Why?
Lines 151-152: Did the author reverse transcribe cDNA? not RNA?
Lines 159-160: The author used the word 'then'. What was the previous operation before resuspending?
Lines 174-178: Were the constructed plasmid used in subsection 2.3? If yes, the plasmid construction and transfection methods should be described before subsection 2.3.
Lines 180-181: Where did the 243 ducks come from? Was this a part of the 500 male ducks described in subsection 2.1?
Lines 181-184: Please specify the RXRA gene id that the author used to design the primer.
Line 193: Please analyze the total feed intake and final body weight as the growth traits.
Lines 194-195: Did the author analyze the carcass trait? If yes, please show the data in the Result section.
Lines 214-215: Please change the name of Figure (B). The present title told 300 uM of oleic acid was used at 0, 2, 4, and 6 days. It differs from those stated in Materials and Methods (Lines 121-126).
Line 216: What was induced in the DPCAs?
Lines 241-245: These explanations should also be written in the Materials and Methods section.
Line 253: Please change oeRXRA to OERXRA.
Lines 271-273: Please specify the gene id that the author used to check the mutant allele in the Materials and Methods section.
Lines 304-305: Please check the result carefully. Eight SNPs were not related to FCR significantly.
Lines 305-307: From Table 4, g.5,952,188 A>T was also related to ADFI significantly.
Lines 310-311: How about the CC genotype of g.5,952,667 C>T?
Table 4: There are 7 SNPs whose genotypes do not sum to 243. If there are any reasons such as genotyping errors, please describe them in the Materials and Methods section. Why is the total number of genotypes for g.5,952,666 G>A 247?
Lines 313-314 and 326-327: The author did not use the superscripts in the Table.
Figure 5: Please use the same SNP name as other data.
Table 5: What are 128, 85, and 16 indicated? If it is a number of the studied duck, the total is 229. What duck population was used for this analysis?
Line 350: Please add the unit after 6.34.
Lines 365-372: The authors should discuss how the structure and role of the RXRA change by a missense mutation. The references (38 and 39) showed the effects of amino acids when supplemented in food, and it is questionable whether the amino acid substitutions in RXRA found in this study would have a similar effect.
Lines 381: What result can conclude that the RXRA gene facilitates fat accumulation via the PPARG-C/EBPA signal? The authors should provide more information about the PPARG-C/EBPA signal.

Author Response

(The authors gave the same response as above.)

Reviewer 3 Report

The authors indicated that retinoic acid X receptor alpha (RXRA) has an important role in adipogenesis in deck adipocytes, and the genetic variation of it associates feed efficiency. It is well known that RXRA is vital to regulate substrate metabolism in mammals, and, they demonstrated that its function is conserved in ducks. Moreover, they found 12 SNPs in RXRA gene associates feed efficiency. The present study indicated some novel findings which will contribute to duck production. I request some modifications as follows.

1.       The authors demonstrated that in duck preadipocytes, RXRA promotes PPARG and C/EBPA transcription and adipogenesis. The expression of genes-related to lipid metabolism is also changed the later stage of cell differentiation??

2.       In materials and methods, they indicated the isolation and cultivation of preadipocytes in ducks. This method is newly developed in this study?? If some previous studies which show the methods, they should state reference.

3.       P13L332-333 The authors state the importance of oleic acid in cell differentiation of duck preadipocyte. This is very important information to readers and I think this information should move to materials and methods.

4.       The present study found the four SNPs in RXRA gene affect amino acids mutation. These mutations affect the binding with transcriptional factor such as PPARG and C/EBPA, and the expression of downstream genes?? If possible, please discuss this point.

5.       The legend of Figure 1. Figure 1 include not only RXRA expression but also fat accumulation. So, authors should change the Fig title more suitable.

Minor point

6.       Rpm/min ->rpm or r/min

7.       Line 253 oeRXRA->OERXRA

Author Response

(The authors gave the same response as above.)

Round 2

Reviewer 1 Report

L 58 ff: Once again. The SNP analysis was not carried out on the cell cultures but on the DNA of male ducks. These are two different things. Just make two sentences out of one paragraph.

L 77: correct (N=3) to (n = 3)

L 82: "The duck preadipocytes isolated in this study showed results: [...]" This is part of the results. Delete this. The second part of the sentence is correct and can stay there.

L 86: It´s the first time you used the abbreviation DPAC in the manuscript (except in the abstract). Give the full term.

L 88: In which machine did you measure the TG content?

L 97: "Pan et al[20]." correct to Pan et al. [20]

Please insert a scale in the pictures of Figures 1B and 2D.

L 190: The first sentence is a heading. Delete this sentence.

L 242: Insert the average values of IBW, FBW, and FI of the n = 243 population or refer to Tables 3 and 4 of your manuscript.

Author Response

Thank you for your valuable comments again.

Our response can be found in the attachment.

Reviewer 2 Report

I thank the authors for improving their papers according to my comments and suggestions. The following are my comments on the author's response and their revised paper.

Comments on the author's response:
Point 10: There is no corresponding information in the revised paper.
Point 16: The information 'after plasmid transfection' in the first sentence may confuse the reader because it does not relate to the subsection title for 2.3 and the contents of previous subsections. How about the following?
The authors could remove the first sentence in subsection 2.3 (DPACs were collected 24 hours after plasmid transfection). And the sentence 'For plasmid transfected DPACs, cells were collected 24 hours after transfection.' could be added at the end of subsection 2.3.
Point 17: How did the author select the 243 ducks from the 500 duck population? Please clearly state in the paper that the 243 ducks used for the SNPs analysis were chosen from 500 male ducks.

Minor comments:
Overall. Please check the SNP name carefully. The 12 SNP names used in the paper sometimes do not include spaces and sometimes do. Also, most SNP names contain alphabetic characters, but sometimes they do not.
Line 35. Please delete 'et al.' after Luiting. One author wrote this publication.
Line 36. Please add 'et al.' after Bai.
Line 55. Please confirm '(RXRA)'. Is it not (PPARG)?
Line 64. The author deleted the detailed information about their animal experiments shown before (Lines 78-92 in the earlier version). It is better to retain this information in the revised paper. Without this information, the initial body weight (IBW, 21d) and final body weight (FBW, 42d) stated on Line 242 may confuse the reader.
Line 98. The author deleted the reference gene and calculation methods information. Please add this information (Lines 153-155 in the previous version) to the modified paper.
Line 174: Please add the forward slash between C and EBPA.
Figure 2C: Please add 'OE' before 'RXRA' on the x-axis.
Line 200: Please add the 'normal' before the 'control group'.
Line 219: The author deleted the definition of wild-type alleles in the modified paper. Please add this information (Lines 271-273 in the earlier version) to the modified paper.
Line 304: Please add the space before the unit 'Mb'.
Lines 316 and 317: Please add the duck population's sample size (n = 243).

Author Response

(The authors gave the same response as above.)
